# The Study of the Performance of the Diamond Wheel's Steel and CFRP Hubs in Tungsten Carbide (WC) Grinding

Yao-Tsung Lin [1] , Kai-Jung Chen [2] , Chun-Yen Chen [1] , You-Xhiang Lin [2] and Ming-Yi Tsai [1],*

1    Graduate Institute of Precision Manufacturing, National Chin-Yi University of Technology, No. 57, Sec. 2, Zhongshan Rd., Taiping Dist., Taichung 41170, Taiwan; train@ncut.edu.tw (Y.-T.L.); jack1baycomtw@gmail.com (C.-Y.C.)
2    Department of Mechanical Engineering, National Chin-Yi University of Technology, No. 57, Sec. 2, Zhongshan Rd., Taiping Dist., Taichung 41170, Taiwan; hskchen5@ncut.edu.tw (K.-J.C.); bobo6202@gmail.com (Y.-X.L.)
*    Correspondence: mytsai@ncut.edu.tw

**Abstract:** Tungsten carbide (WC) has been widely utilized in recent years in the hardware, mechanical, and chemical industries and in national defense because of its high hardness, anti-wear, low temperature, and anti-corrosion properties. However, using it for grinding is also challenging because the WC material has high hardness and brittle characteristics. The typical hub of a diamond wheel is made of steel. In high-speed grinding, the steel hub of the diamond wheel is subjected to gravity and centrifugal forces, which cause grinding wheel vibration, poor workpiece processing quality, and a short machine life. Therefore, this study used a carbon-fiber-reinforced thermoplastic (CFRP) hub to replace the steel hub when grinding the WC workpiece. It aimed to investigate methods to reduce oscillation, improve chip efficiency, and increase accuracy in the WC workpiece. The research results demonstrated that using a CFRP hub in the grinding wheel can reduce the oscillation when the peripheral speed of the grinding wheel is at 20–100 m/s. Additionally, the surface roughness average (Ra) of the workpiece can be reduced to 3.2–25.4% and the ten-point height of irregularities (Rz) can be reduced to 18.9–44% compared to using a steel hub in the grinding wheel.

**Keywords:** tungsten carbide (WC); hardware; steel hub; CFRP hub; diamond wheel

## 1. Introduction

Tungsten carbide (WC) and SiC are hard and brittle materials. The material of the hardness is over HRC 70. [1] WC has been widely used in manufacturing for various applications, including molds and dyes, cutting tools, wear-resistant components, and coatings. WC often faces problems with the grinding wheel's wear and brittleness and the roughness of the surface of the workpiece during the grinding process because of its material behavior. As a result, this material behavior makes the grinding process difficult [2–6]. Thus, developing a novel method to improve its efficiency and reduce the product's surface roughness is an excellent contribution to scientific research.

In recent years, many researchers have demonstrated various methods for high-speed grinding that can deal with hard and brittle materials to improve efficiency [7,8]. Li P. et al. [3] investigated methods to achieve high-efficiency and high-quality grinding technology in the precision chip process of brittle glass–ceramic material. Their findings revealed that using multi-step high-speed grinding technology helps achieve high-efficiency and high-quality grinding for glass ceramics. Yang L. et al. [9] noted that high-speed grinding exceeds the peripheral speed of the grinding wheel by 80 m/s. The precision and surface integrity of the workpiece with more difficult-to-process materials can be improved during high-speed grinding [10]. The surface and sub-surface of the workpiece are capable of manufacturing molds for optical elements with great form precision. Gu K. K. et al. [11] claimed that increasing the grinding speed can improve the removal rate, and a smooth

surface can be obtained. Ren Y. H. et al. [12] presented the ultra-high-speed grinding technology and asserted that it could reduce the temperature of the grinding point during the grinding process. Overall, all the studies stated that high-speed grinding has a good performance on the surface roughness of the workpiece. Notably, the hub's grinding wheel material selection remains related to the geometric accuracy, stiffness, thermal deformation, motion stability, and vibration resistance.

Traditionally, the hub's grinding wheel is made of steel material. However, a steel grinding wheel is only suitable for peripheral speeds below 60 m/s. It produces deflection when the peripheral speed of the diamond wheel exceeds 60 m/s due to the steel grinding wheel's gravity, rotational centrifugal force, and thermal expansion. These factors reduce the accuracy of hard and brittle materials and speed up the grinding wheel wear [13]. In addition, the processing machine generates wear, heat, and power consumption, which shortens the life of the spindle due to centrifugal force and gravity. Overall, the grinding wheel of the steel hub can only be used in low-speed grinding applications. On the other hand, some studies changed the material of the hub to carbon-fiber-reinforced polymer (CFRP) and compared its advantages and disadvantages on the grinding performance. Kumekawa N. et al. [14] presented CFRP, which has lighter characteristics, higher strength, and a lower thermal expansion coefficient compared to steel. Tawakoli T. et al. [15] argued that damping affects the vibration and chip thickness of the workpiece within the grinding wheel and during the grinding process. The CFRP hub can produce a high damping capability to stabilize the vibration during high-speed grinding. Yang L. et al. [13] performed a finite element analysis to analyze the stress, strain, damping characteristics, and thermal performance. The results indicated that the CFRP hub has uniform stress and deformation distribution when the CFRP material weaves and laminates at 0°, 45°, and 90° angles, implying that the CFRP hub has better dynamic characteristics. Li W. et al. [16] explained that the main causes of residual stress on the workpiece surface are mechanical deformation, thermal expansion and contraction, and material phase change. Thus, Kizaki T. et al. [17] examined the hub of the grinding wheel by combining a novel hub with steel and a CFRP substrate to perform high-speed grinding. The results revealed that the inertia reduced by 61.4% more than a conventional grinding wheel with a steel hub when the CFRP hub of the grinding wheel rotated at a high peripheral speed. Thermal expansion is also reduced by 59% due to centrifugal force. In addition, the hub that combines the steel and CFRP substrate has a different thermal expansion coefficient. Therefore, it can lead to the distortion of the grinding wheel under high-speed grinding, possibly resulting in unstable oscillation during high-speed grinding. It is essential to define the usage scenarios of the CFRP hub grinding wheel to effectively reduce the generation of negative benefits, such as heat and vibration effects.

This research aims to compare the performance of the steel and the CFRP hub grinding wheels to define the latter's usage scenarios. The performance evaluation was constructed with a knock-test method for the vibration effect and the process verification under different peripheral speeds (at 20, 40, 60, 80, and 100 m/s) for surface roughness performance and grinding wheel life evaluation. In addition, this study aims to develop a carbon-fiber-reinforced thermoplastic (CFRP) hub for high-speed grinding. This new hub is expected to be lighter and less deformable under high-speed rotation and thus able to obtain a higher material removal rate, smoother workpiece surface, and longer tool life. This novel innovation can help the CFRP hub grinding wheel application in manufacturing and reduce the grinding challenge in tungsten carbide.

The key highlight of this study is the comprehensive comparison between the steel and the CFRP hub grinding wheels to determine the most suitable applications for the CFRP hub. First, the study employs a knock-test method to assess the vibration impact and conducts process verification at various peripheral speeds (ranging from 20 to 100 m/s). This evaluation encompasses surface roughness performance and grinding wheel longevity, providing valuable insights into how the two hub materials perform under different conditions. Secondly, this research aims to introduce an innovative carbon-fiber-reinforced

thermoplastic (CFRP) hub designed specifically for high-speed grinding applications. This new hub is engineered to be lightweight and less prone to deformation during high-speed rotation. The expected outcomes of this innovation are threefold: a higher material removal rate, smoother workpiece surfaces, and an extended tool life. Moreover, the study carries significant implications for the manufacturing industry by offering a solution that enhances grinding wheel performance in high-speed applications, particularly when dealing with challenging materials like tungsten carbide. The CFRP hub grinding wheel's potential to address these challenges can improve the manufacturing efficiency and product quality. In summary, this study's core contribution lies in exploring the CFRP hub's suitability for high-speed grinding, its innovative hub design, and its potential to provide practical solutions to manufacturing challenges, particularly in tungsten carbide processing.

## 2. Experimental Method

### 2.1. Principle Description

The grinding wheel chips through the workpiece material as the workpiece passes underneath. Normal and tangential forces are generated between the grinding wheel and the workpiece, as seen in Figure 1. The forces cause the abrasive grains of the grinding wheel to penetrate the workpiece. Rubbing, ploughing, and cutting are the three stages of material removal. The extent of each stage strongly depends on the physical characteristics of the workpiece, its deformation characteristics, and its reactivity to the abrasive grains and the environment [18].

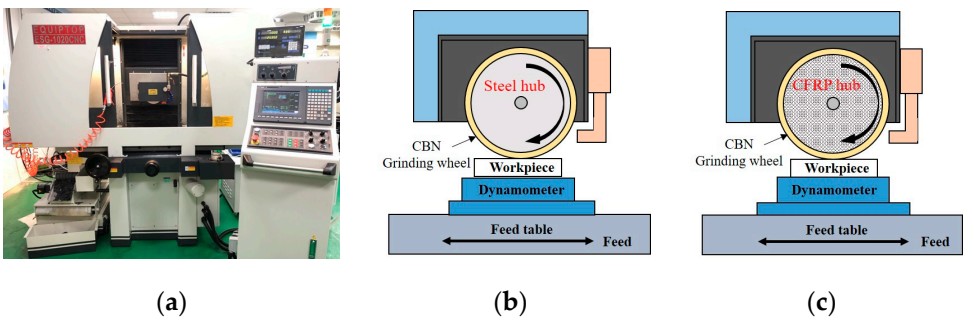

**(a)**　　　　　　　　**(b)**　　　　　　　　**(c)**

**Figure 1.** Precision grinding machine and auxiliary facilities. (**a**) ESG-1020CNC Grinding Machine. (**b**) The diamond wheel of the steel hub. (**c**) The diamond wheel of the CFRP hub.

**Grinding Force.** The normal grinding force acting upon the grits increases due to friction, workpiece shear fracture, and elastic and plastic deformation under grits. The normal grinding force operating on the cutting grits is given by the normal force on the sliding indenter and can be approximated by the following expression [13,14].

$$F_n = H_w A_{X\text{-}Z} \tag{1}$$

$F_n$ is the normal grinding force.
$H_w$ is the material hardness of the workpiece.
$A_{X\text{-}Z}$ is the cross-sectional area of the wheel grit on the workpiece surface.
The dynamometer is used to ensure that grinding forces are correctly identified. The tangential force $F_t$ is the vertical force $F_v$, and the normal force $F_n$ is the horizontal force $F_h$ measured perpendicular to the contact point.

$$F_t = F_h \cos\theta - F_v \sin\theta \tag{2}$$

**Chip formation force.** The specific chip formation energy $U_{ch}$ is divided into the static specific chip formation energy $U_s$ and the dynamic specific chip formation energy $U_d$ [18].

The specific chip formation energy $U_{ch}$ is calculated using the following equation:

$$U_{ch} = \frac{F_{t,ch}V_s}{V_w a_p b} = U_s + U_d. \tag{3}$$

$V_s$ is the grinding wheel velocity.
$V_w$ is the workpiece feed velocity.
$a_p$ is the grinding depth.
$b$ is the grinding width.

The static specific chip formation energy $U_s$ is a constant, which is determined by an experiment based on the element material and the grinding wheel material. On the other hand, the dynamic specific chip formation energy $U_d$ is determined by the element material, grinding wheel material, and grinding processing parameters.

### 2.2. Experimental Equipment

The grinding facilities have a high-speed precision grinder model ESG-1020CNC, meticulously crafted by Equip Top Hitech Corp. Taichung, Taiwan. The operational framework of the grinding facilities includes a schematic delineation depicting the utilization of a diamond wheel installed on both steel and CFRP (carbon-fiber-reinforced polymer) hubs for the execution of the grinding procedure, as showcased in Figure 1. Supplementary to this, Figures 2 and 3, respectively, portray auxiliary components of a static analysis and a process performance evaluation. A comprehensive exposition of the dimensional specifications of the grinding wheel, incorporating steel and CFRP hubs and a measuring apparatus such as WC, is explained in Table 1. The system's configuration parameters are orchestrated to discern the vibrational phenomena and capture distinct morphological transformations induced during the grinding process involving WC. The dimensions of the WC grinding test block are stipulated at $100 \times 60 \times 20$ mm$^3$, with the WC material characterized by a hardness range of 90 to 95 (HRA). A dressing block is employed to refine the grinding wheel's surface to optimize its grinding performance. This process serves to ensure the consistency and efficacy of the grinding operation. The instrumentation used for measuring vibrations entails a non-magnetic dynamometer, a deliberate choice aimed at mitigating any undesirable influence of magnetic forces on the precision and quality of the grinding process.

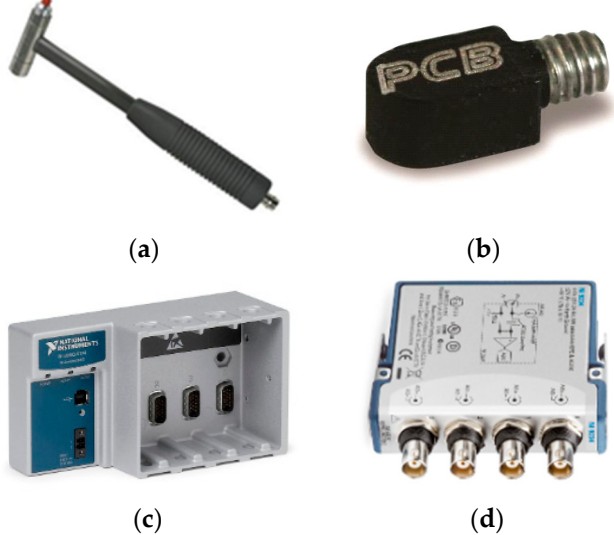

(a) (b)

(c) (d)

**Figure 2.** The auxiliary facilities of the static analysis. (**a**) Impact hammer. (**b**) Accelerometer (Model: 352C23). (**c**) Data acquisition platform built (Model: Card/NI-9174). (**d**) Sound and vibration input module (Model: NI-9234).

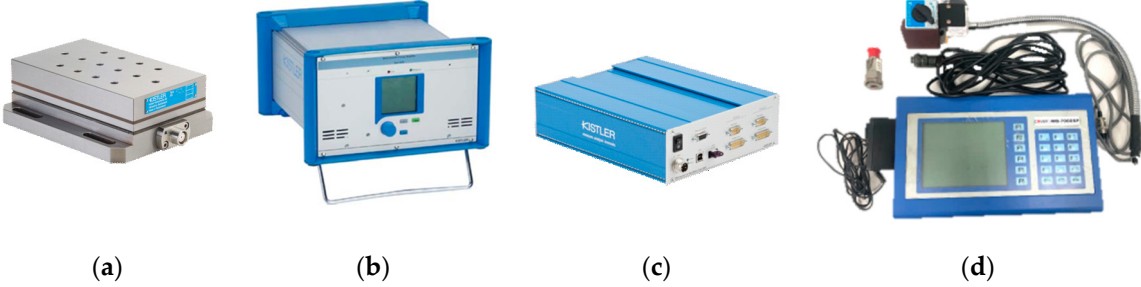

|        |        |        |        |
|:------:|:------:|:------:|:------:|
| (**a**) | (**b**) | (**c**) | (**d**) |

**Figure 3.** The auxiliary facilities of the dynamic analysis. (**a**) Dynamometer (Kistler-9257B). (**b**) Charge amplifier (Kistler-5070A). (**c**) Data capture (Kistler-5697A). (**d**) Dynamic balance calibrator.

**Table 1.** Grinding and dressing parameters.

|                           |                                                                                                   |
|---------------------------|---------------------------------------------------------------------------------------------------|
| Grinding machine          | Equiptop/ESG-1020CNC.<br>Table 250 × 500 mm².<br>Feed speed 10 m/s.                               |
| Grinding wheels           | Ø180 mm × 13 mm × 31.75 mm.                                                                        |
| Hub material              | 1. CFRP.<br>2. Steel.                                                                              |
| Grinding parameters       | The peripheral speeds of the grinding wheel are 20, 40, 60, 80, and 100 m/s.<br>Grinding depth 1 μm. |
| Workpiece                 | WC (Tungsten carbide) 100 × 60 × 20 m³.<br>Hardness 90~95(HRA).                                    |
| Optical Microscope (OM)   | Keyence/VHX-5000/Pixel 1600 (H) × 1200 (V).                                                        |

The selection of abrasive materials for grinding wheels includes attributes such as robust wear resistance, inherent toughness, and elevated hardness. These attributes are paramount when opting for distinct abrasive materials tailored to the varied properties of the workpiece materials. The bonding agent employed in grinding wheel fabrication is predominantly categorized into the following classifications: metal, ceramic, rubber, and resin. The resin bond, distinguished by its heightened elasticity, mitigates impacts and augments buffering capabilities throughout the grinding operation. Notably, its influence on the workpiece surface is minimal due to the reduced normal force resistance exhibited during grinding. This technology is ubiquitous in high-speed grinding wheel configurations [19]. Regarding the abrasive constituents, conventional selections encompass WA (White Aluminum Oxide), GC (Green Silicon Carbide), diamond, and CBN (Cubic Boron Nitride). Notably, diamond abrasives are extensively employed in machining recalcitrant materials characterized by their elevated hardness [20]. Furthermore, diamond abrasives' exceptional heat dissipation properties contribute to maintaining optimal thermal conditions during the grinding process [21]. In the formulation of the grinding wheel under study, diamond abrasives with a Knoop hardness of 7000 and an average particle dimension of 105 μm are synergistically combined with a resin bond.

*2.3. Comparison of the Vibration Effect between the Steel and CFRP Grinding Wheel Hub*

The design and material of the grinding wheel hub determine its static and dynamic states. These properties affect its grinding efficiency and performance. Tawakoli T. et al. [15] suggested that the steel hub of the grinding wheel produces higher amplitude and frequency compared to the CFRP hub when the peripheral speed of the grinding wheel exceeds 100 mm/s. This research used different levels of the hammer's hardness to test the vibration frequency and determine the resonance frequency. Figure 4 illustrates the system facilities. The red arrow indicates the tapping position, while the red circle highlights the dynamometer sensor used to sense the vibration frequency.

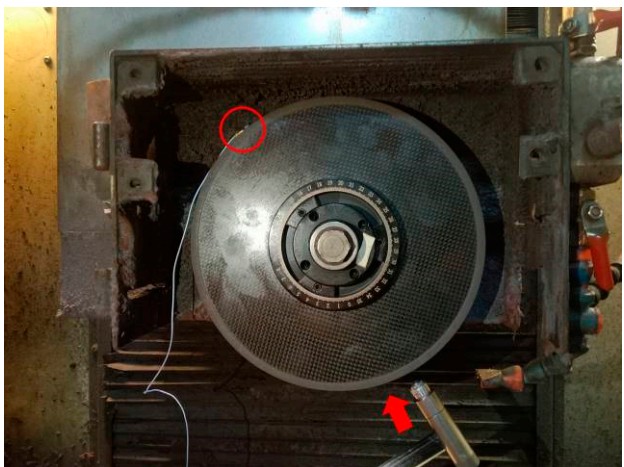 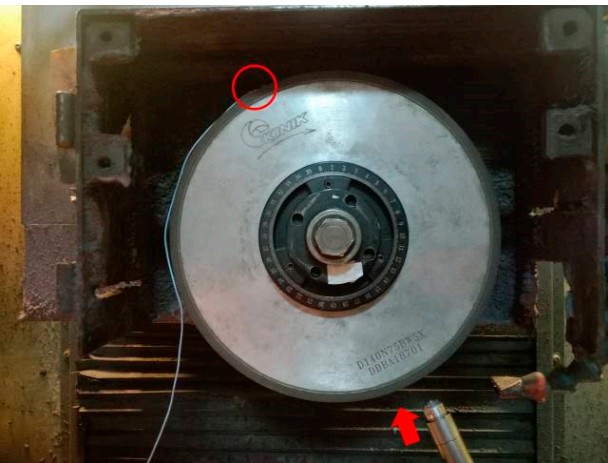

**Figure 4.** Schematic diagram of the knocking test for the grinding wheel of the CFRP hub and steel hub.

Due to the resistance, the body will lead to energy loss when it is in oscillation. As the resistance weakens the energy of the system constantly, the energy of the oscillation decreases steadily. Then, the oscillation will finally stop. This mechanism is called the damping effect [22]. This study used a dynamic balance instrument from Covers Plus International Co., Ltd.—WB-7000SP—(Taipei, Taiwan) to measure the vibration of the steel and CFRP hubs when the grinding wheel was in a static state, as shown in Figure 4.

*2.4. Comparison of the Surface Performance between Steel and CFRP*

The workpiece exhibits a subtle degree of surface roughness, characterized by an inherent smoothness that attests to its superior quality. In this context, the investigation employs the arithmetic mean roughness (Ra) and the ten-point mean roughness (Rz) as metrics to quantify the surface roughness of the workpiece following the grinding procedure. To ascertain both the surface roughness and the topographical features of the WC workpiece, the experimental apparatus integrates the Performance NewView8300 white light interferometer with a digital microscope. The results of each measuring surface area included ten repeated measurements and calculated the mean and standard deviation.

**3. Experimental Results**

*3.1. The Oscillation of the CFRP Hub and the Steel Hub Grinding Wheel in the Grinding Process*

This study used a static state to evaluate the different vibrations that emerge when an external force touches the grinding wheel of the CFRP hub and the steel hub. In particular, it used soft and hard hammers to knock the diamond wheel to produce higher, middle, and lower vibration frequencies. The higher frequency was defined as 5500 Hz, the middle frequency was defined as 500 Hz, and the lower frequency was defined as 300 Hz. The goal was to investigate the vibration in the CFRP hub and the steel hub grinding wheel within a certain period in the static test. The experimental results revealed that the initial amplitude of the CFRP hub grinding wheel exceeds that of the steel hub, as seen in Figure 5(a1,b1). However, the amplitude of the steel hub grinding wheel is slightly over that of the CFRP hub when the vibration time exceeds 0.05 s. The experimental findings show that the CFRP hub grinding wheel is more significant than the steel hub in the initial amplitude value. However, the CFRP hub of the grinding wheel can stabilize the vibration when an external force strikes the grinding wheel at a low frequency.

The middle frequency was defined when the tapping frequency reached 500 Hz. The initial amplitude of the CFRP hub grinding wheel is smaller than that of the steel hub and can stabilize the oscillation at 0.1 s, as seen in Figure 5(a2,b2). The chatter vibration of the CFRP hub grinding wheel is also more moderate than that of the steel hub grinding wheel when the time is lower than 0.05 s.

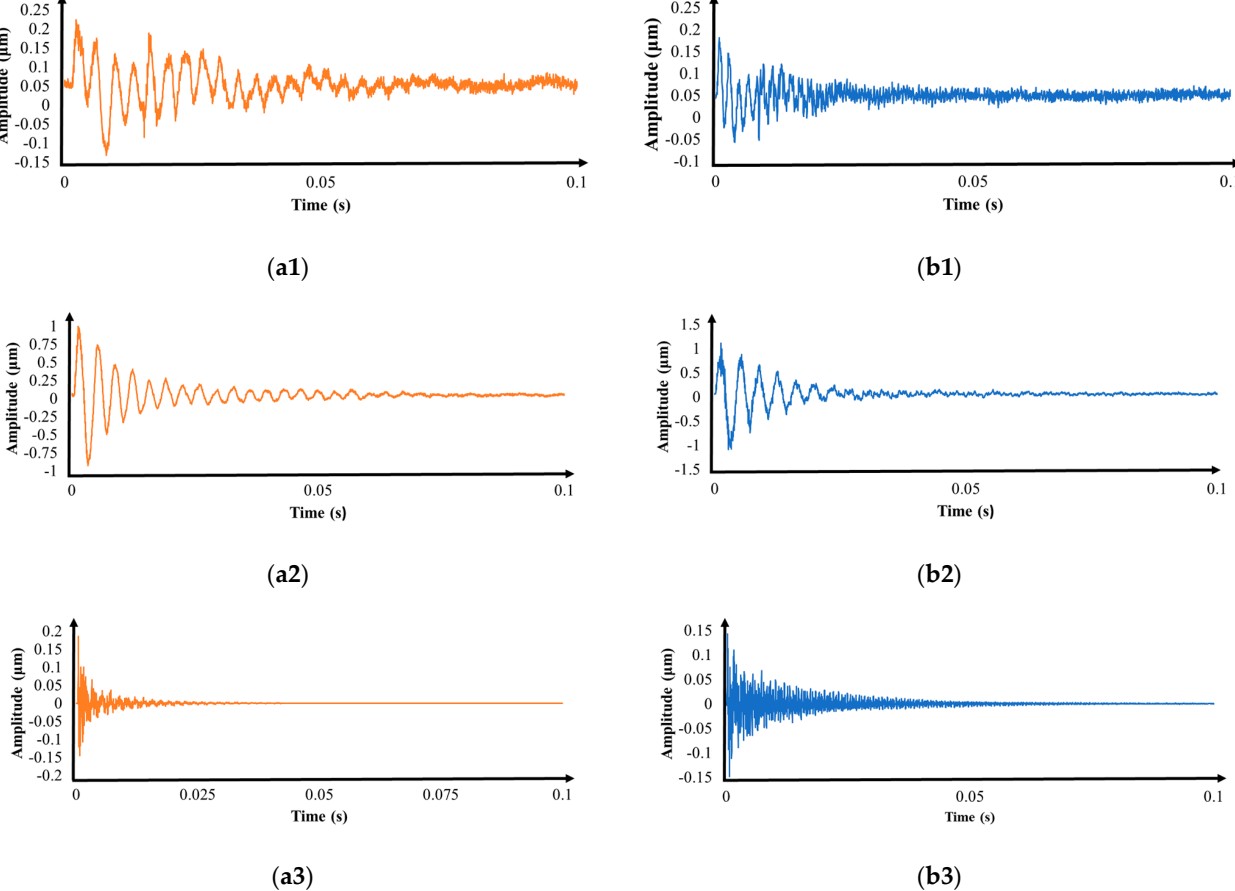

**Figure 5.** Relationship of amplitude and time when the grinding wheel of the CFRP and steel hubs are tapping at different frequencies. (**a1**) The grinding wheel of CFRP hub vibration at 300 Hz. (**a2**) The grinding wheel of CFRP hub vibration at 500 Hz. (**a3**) The grinding wheel of CFRP hub vibration at 5500 Hz. (**b1**) The grinding wheel of steel hub vibration at 300 Hz. (**b2**) The grinding wheel of steel hub vibration at 500 Hz. (**b3**) The grinding wheel of steel hub vibration at 5500 Hz.

The CFRP hub and the steel hub grinding wheel amplitude values are quite similar when the tapping frequency reaches 5500 Hz. However, the CFRP hub grinding wheel can stabilize the oscillation time quickly, ranging between 0.025 and 0.05 s. The opposite happens when the steel hub grinding wheel exceeds 0.075 s, and the chatter vibration becomes more obvious than that of the CFRP hub grinding wheel. The experimental results are shown in Figure 5(a3,b3).

The abovementioned experimental results suggest that high-frequency vibration can be easily produced when the peripheral speed of the grinding wheel increases, and the grinding wheel grinds a workpiece with extreme hardness. The CFRP substrate has better damping ability for grinding the performance products [11]. Meanwhile, the grinding wheel of the steel hub can produce a chatter vibration and a larger amplitude to influence the quality of the grinding process.

### 3.2. The Surface Roughness and Morphology of a WC Workpiece after Grinding

Figures 6–8 illustrate the surface roughness average (Ra) and ten-point height of irregularities (Rz) and morphology after the grinding wheel of the steel hub and CFRP hub grind the WC workpiece at 1.0 μm. The peripheral speed of the diamond wheel was set at 20, 40, 60, 80, and 100 m/s. The measurement facilities used a CMOS image sensor with virtual pixels at 1600 (H) × 1200 (V). The surface roughness Ra decreased to 0.122 μm from 0.399 μm, while the Rz decreased to 1.834 μm from 4.312 μm when the peripheral speed of the steel hub grinding wheel increased from 20 m/s to 100 m/s, as shown in Figure 5. The surface roughness Ra

decreased to 0.091 µm from 0.386 µm, while the Rz decreased to 1.027 µm from 3.495 µm when the peripheral speed of the CFRP hub grinding wheel increased from 20 m/s to 100 m/s. This finding means that the surface fineness is better for the CFRP hub than the steel hub. In addition, the surface morphology of the workpiece was also decreased in the values of Sa, Sq, and Sz, with the peripheral speed of the grinding wheel increasing from 20 m/s to 80 m/s, as shown in Figures 7 and 8. These findings show that the diamond grinding wheel of the CFRP hub is suitable for grinding hard and brittle WC materials at high speeds.

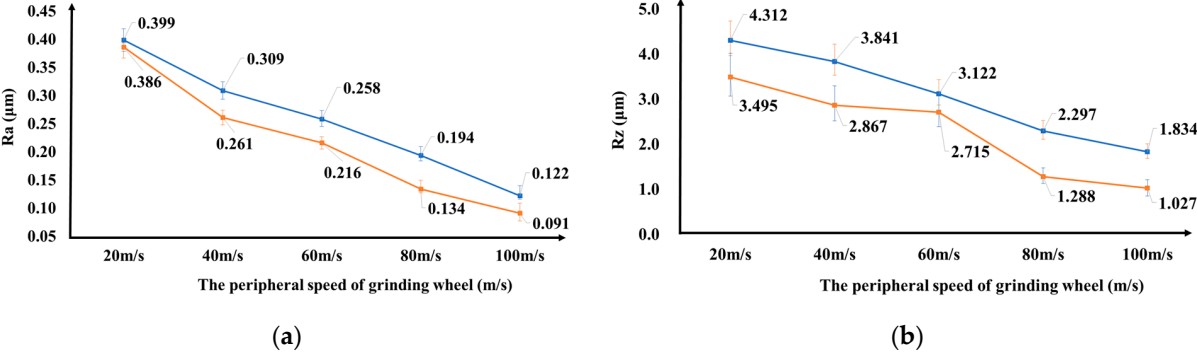

(**a**)          (**b**)

**Figure 6.** The surface roughness in Ra and Rz after the diamond wheel of the CFRP hub (orange) and steel hub grind (blue) at 1.0 µm in the WC workpiece. (**a**) represent the relationship in Ra and the peripheral speed of grinding wheel. (**b**) represent the relationship in Rz and the peripheral speed of grinding wheel.

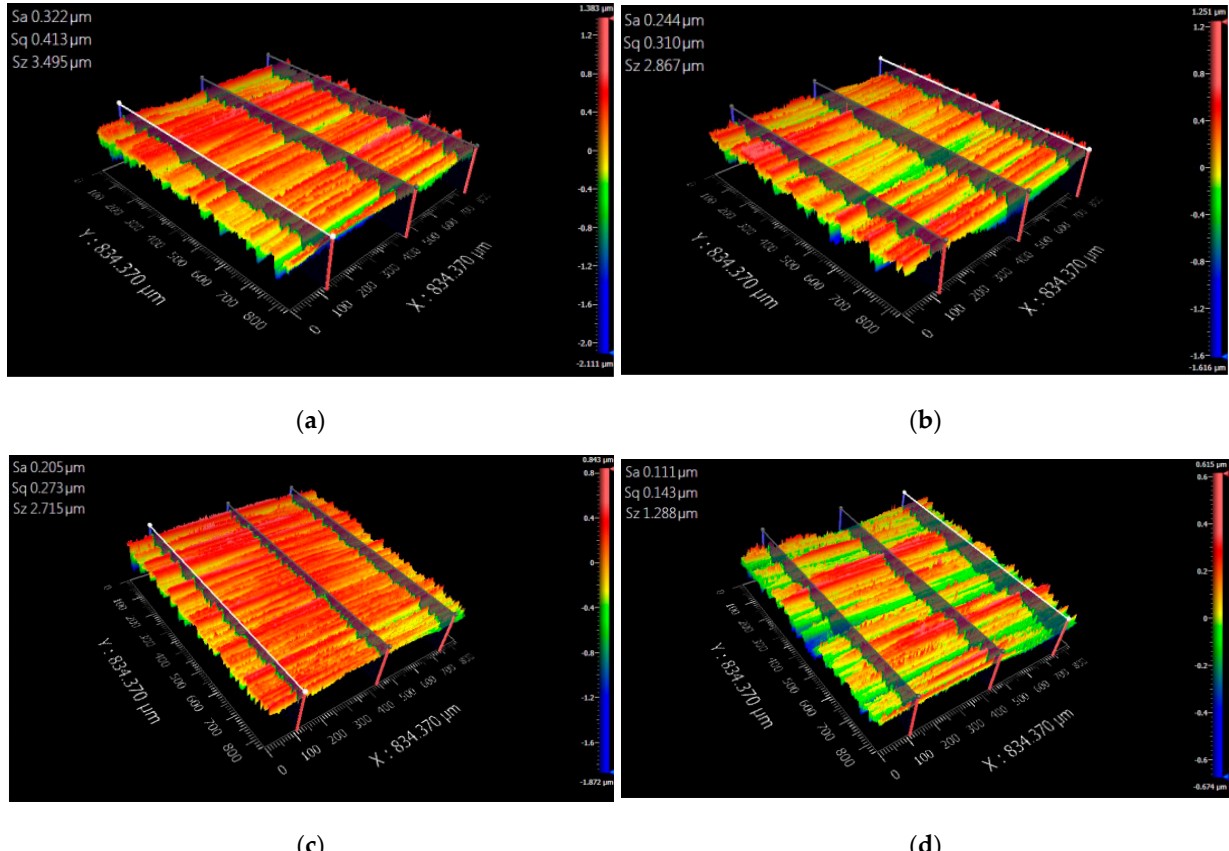

(**c**)          (**d**)

**Figure 7.** The surface morphology when the diamond wheel of the CFRP hub is grinding at 1.0µm in the WC workpiece. (**a**) The peripheral speed of the grinding wheel at 20 m/s. (**b**) The peripheral speed of the grinding wheel at 40 m/s. (**c**) The peripheral speed of the grinding wheel at 60 m/s. (**d**) The peripheral speed of the grinding wheel at 80 m/s.

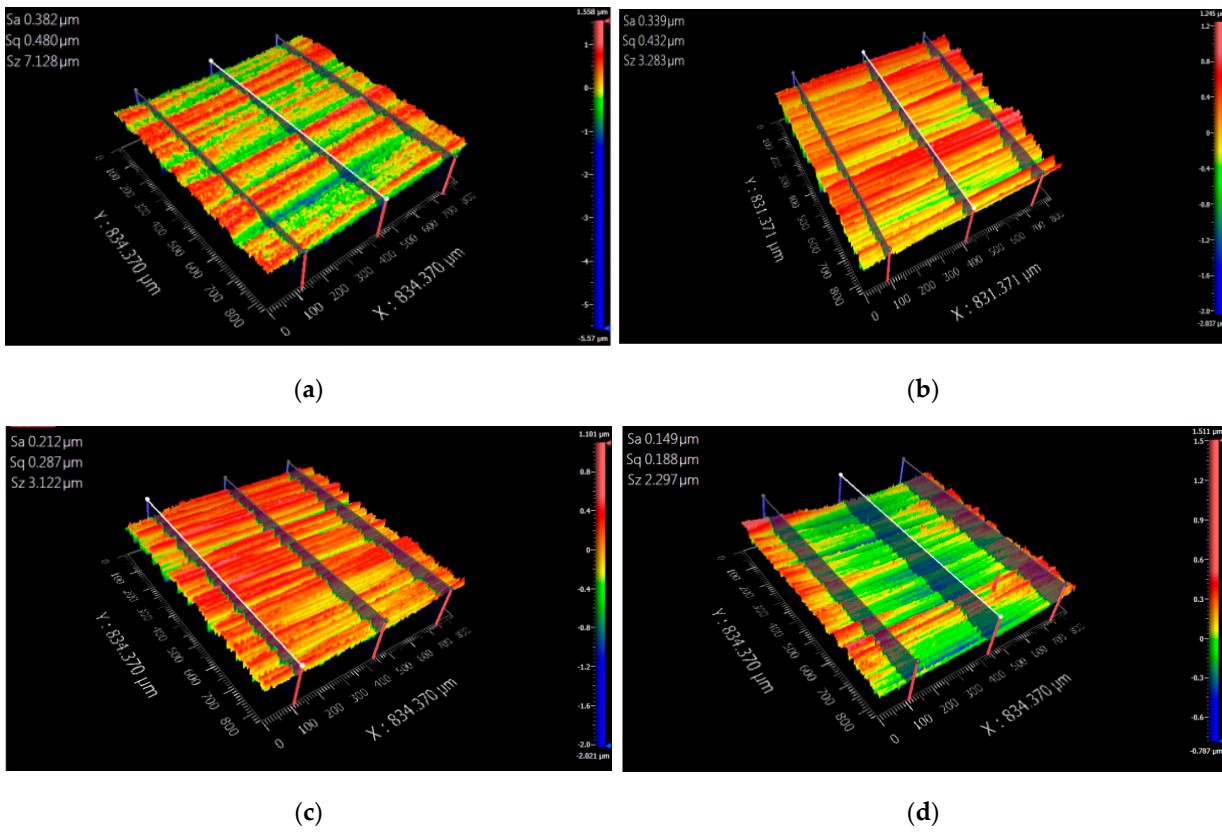

**Figure 8.** The surface morphology when the diamond wheel of the steel hub is grinding at 1.0 μm in the WC workpiece. (**a**) The peripheral speed of the grinding wheel at 20 m/s. (**b**) The peripheral speed of the grinding wheel at 40 m/s. (**c**) The peripheral speed of the grinding wheel at 60 m/s. (**d**) The peripheral speed of the grinding wheel at 80 m/s.

## 4. Discussion

The material removal process involving a grinding wheel encompasses distinct stages: rubbing, plowing, and cutting. Each stage is pivotal in shaping the intricate interplay between abrasive grains and the workpiece surface [17]. The initial rubbing phase introduces abrasive grains to the workpiece surface, initiating the abrasion process that gradually removes material through contact. This contact simultaneously triggers elastic deformation and generates heat through friction within the system. Significantly, the resultant effects induce vibrational patterns in the grinding wheel, highlighting the dynamic nature of the process. As the grinding wheel's vibrations amplify the effects of elastic deformation and heat generation, abrasive grains embedded in the workpiece material give rise to plastic deformation. Consequently, the workpiece surface experiences a subtle transformation characterized by the emergence of subsurface stresses and the chipping of abrasive grains into the material matrix. This embedding phenomenon triggers the migration of workpiece material, compelling it to shift toward the front and sides of the abrasive grains, culminating in the formation of distinctive grooves—a phenomenon termed 'the plowing stage'. This phase accentuates the critical role of abrasive grains in inducing plastic deformation and redistributing material. However, the plowing stage introduces challenges, primarily centered around the extrusion of abrasive grains against the workpiece, which leads to heightened friction and intensified heat generation. In addition, elevated temperatures become a focal concern during this phase, potentially affecting both the grinding process' efficiency and the workpiece's quality. Reducing the vibration during the grinding process can effectively reduce the impact of heat. Therefore, reducing the quality of the grinding wheel and the vibration during the machining process can reduce the surface roughness. If the abrasive grains persist in interacting with the workpiece, the conditions conducive to

shear and slip will eventually emerge, resulting in the commencement of the cutting stage. In this phase, shear forces and slip motions intertwine, facilitating the formation of chips in the workpiece material. The formation of these chips represents a fundamental component of the grinding process, directly influencing the extent of material removal and the resultant surface finish quality. Throughout the grinding operation, the presence of grooves or swells significantly affects the ultimate roughness of the ground surface, considerably affecting the final product. The issue of vibration during the grinding process emerges as a pivotal consideration in optimizing ground surface quality. Effectively addressing and minimizing vibration tendencies can substantially reduce the roughness values on the finished surface, ultimately enhancing the precision and overall quality of the grinding operation.

Based on the oscillation results, when an external force displaces the grinding wheel from its equilibrium position, it induces free vibrations. These vibrations vary depending on the material properties of the hammers striking the diamond wheel, resulting in different external forces and corresponding frequencies. As the material properties of the hammer become stiffer, the external force increases. Figure 5 illustrates a noteworthy finding: the CFRP hub, upon being displaced, returns to its equilibrium position roughly 50% faster than the steel hub. However, it is important to note that the maximum vibration amplitude of the CFRP hub reaches 0.2 um, which is greater than the 0.15 um observed for the steel hub under high vibration frequencies, specifically at 5500 Hz, where the external force is substantial. The reduction in the vibration amplitude of the CFRP hub during free vibrations can be attributed to its superior damping ability compared to that of steel. Notably, the CFRP hub comprises two distinct materials—carbon fiber fabrics and a resin matrix, arranged in a layered structure. These structural differences set the CFRP hub apart from a single-material hub like the steel hub and allow it to absorb vibrations effectively. This outcome can be attributed to the CFRP hub's lower density and elastic modulus, which render it lighter and more conducive to shorter oscillation times. Furthermore, the CFRP hub demonstrates enhanced damping properties, effectively mitigating resonance frequencies and reducing oscillation times during grinding. This observation aligns with the findings of Yang L. et al. [13], reaffirming the significance of hub material properties in shaping grinding dynamics. A critical aspect emphasized by the study is the influence of the hub material on the grinding process' longevity and operational stability. The substantial disparity in material density between the steel and CFRP hubs results in distinct thermal fatigue and wear behaviors. Specifically, the higher density of the steel hub can exacerbate thermal fatigue damage in the spindle or wear in the grinding machine's bearing. The subsequent reduction in spindle life has direct implications for power efficiency. Introducing a CFRP hub addresses this challenge by stabilizing the grinding process swiftly and introducing smaller amplitudes in high-speed grinding.

Figure 6 displays the measured ground surface roughness, revealing an intriguing comparison between the two materials' hubs. The transition from a peripheral speed of 20 to 100 m/s is associated with a notable improvement in surface smoothness. Specifically, when the steel hub of the grinding wheel grinds at a peripheral speed of 20 m/s, the Rz/Ra ratio stands at 10.8 in the workpiece surface roughness. Simultaneously, the CFRP hub of the grinding wheel, under the same peripheral speed of 20 m/s, records a ratio of 10.2. These values are remarkably similar. However, when the peripheral speed of the steel hub grinding wheel escalates to 100 m/s during the grinding of the WC workpiece, the Rz/Ra ratio surges to 15.03. Conversely, after the CFRP hub of the grinding wheel processes the WC workpiece, the Rz/Ra ratio only reaches 11.29, as indicated in Table 2.

These outcomes suggest that changes influence the grinding wheel with a CFRP hub in peripheral speed less. In other words, the CFRP hub is better suited for high-speed grinding wheel configurations. Furthermore, the results highlight that the performance of the CFRP grinding wheel exhibits an optimal boundary under usage conditions, typically at a peripheral speed of around 60 m/s. At this speed, the Rz/Ra ratio in the CFRP hub is approximately 4.1, notably smaller than ratios at other speeds. This implies that the CFRP hub performs most effectively at a peripheral speed of roughly 60 m/s, which is the same

as the critical speed of high-speed grinding. In contrast, the performance of the steel hub diminishes as peripheral speed increases. In summary, these observations underscore the suitability of the WC material for high-speed machining due to its hardness and brittleness. This finding elucidates how the CFRP hub of the diamond wheel excels in stabilizing the grinding process, mainly when dealing with tough and brittle WC material.

**Table 2.** Comparison of the characteristics of the CFRP hub and steel hub.

| | Weight | Damping | Rz/Ra at the Peripheral Speed of 20 m/s | Rz/Ra at the Peripheral Speed of 80 m/s |
|---|---|---|---|---|
| CFRP hub | Lighter | Lower | 10.2 | 11.29 |
| Steel hub | Heavy | Higher | 10.8 | 15.03 |

It is intriguing that despite working under identical grinding conditions, the grinding wheel with a CFRP hub consistently delivers a finer surface roughness. However, this phenomenon remains inadequately explained when considering factors like the grinding wheel's weight, the material properties of the hub, and the abrasive characteristics alone. To delve deeper into this phenomenon, a visual examination of the surfaces generated by both material hubs was conducted using Zygo NewView8000. Figures 7 and 8 clearly depict the striking differences between the surfaces created by these two distinct material hubs, especially when subjected to varying peripheral speeds. The surfaces produced by the CFRP hub exhibit minimal height variations, resulting in a smaller Rz value than those generated by the steel hub. This visual evidence underscores the pivotal role of the hub material in shaping surface quality. It underscores the unique advantages of the CFRP hub in achieving superior surface finishes across a range of grinding conditions. Furthermore, it is worth noting that there are more pronounced changes in surface elevation when the steel hub is used at high peripheral speeds of the grinding wheel. These heightened changes can be attributed to the high-frequency vibrations generated within the grinding wheel, particularly when equipped with the steel hub material. These vibrations contribute to a more brittle material removal process, further underscoring the hub material's importance in influencing the grinding outcome.

## 5. Conclusions and Future Work

In conclusion, the study elucidates the capacity of a CFRP diamond wheel hub to counteract vibration frequencies and enhance surface fineness in WC workpiece materials compared to a steel hub. This performance enhancement is attributed to the CFRP hub's unique material properties, encompassing a lower density, a higher rigidity, a lower thermal expansion coefficient, and superior damping when juxtaposed with the steel grinding wheel hub. The implications are profound, with potential applications in high-speed grinding to bolster both efficiency and surface quality for WC material. Future research avenues are suggested, including using materials with enhanced hardness and brittleness to validate the CFRP hub's reliability when grinding challenging materials. The ultimate goal is to input experimental parameters into a comprehensive database systematically, optimizing machining databases for notoriously hard, brittle, and challenging-to-cut materials, thus furnishing the grinding industry with valuable references. Thermal generation and cooling efficiency can also be investigated further to refine high-speed grinding technologies. Moreover, this study underscores the multidimensional nature of the grinding process, combining material science, engineering dynamics, and operational considerations to yield precision and efficiency enhancements.

**Author Contributions:** Investigation, C.-Y.C.; Resources, Y.-X.L.; Data curation, K.-J.C.; Writing—review & editing, Y.-T.L.; Project administration, K.-J.C. and M.-Y.T. All authors have read and agreed to the published version of the manuscript.

**Funding:** This research was funded by KINIK COMPANY, MOST 111-2218-E-167-001 and MOST 110-2221-E-167-016-MY3.

**Data Availability Statement:** Data are contained within the article.

**Conflicts of Interest:** The authors declare that this study received funding from KINIK COMPANY. The funder was not involved in the study design, collection, analysis, interpretation of data, the writing of this article or the decision to submit it for publication.

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
