# Peer review of "The Study of the Performance of the Diamond Wheel’s Steel and CFRP Hubs in Tungsten Carbide (WC) Grinding"

_applsci, doi:10.3390/app132212131_

Round 1
Reviewer 1 Report (New Reviewer)
Comments and Suggestions for Authors
Dear Authors,
The manuscript presented by authors has few things to be addressed before it is ready to be published. Below are my questions.
1. Please explain the novelty in the introduction clearly.
2. Please compare your studies with existing carbide materials on the basis of their performance.
3. In Figure 5 Please make the amplitude scale same for better comparison.
Comments on the Quality of English Language
Quality of English can be further improved.
Author Response
1-1. Please explain the novelty in the introduction clearly.
Reply 1-1.We explain in the para 4 of the introduction.
This research mainly used CFRP hub and steel hub of the grinding wheel to compare the performance in the surface roughness for the carbide materials(WC). As WC is one of hardness-brittle material. It must use the novel hub to increase the performance in the surface roughness. Steel hub is the traditional grinding wheel. The experimental results show the performance of the CFRP hub is better than the steel hub in the grinding WC materials.
1-2. Please compare your studies with existing carbide materials on the basis of their performance.
Reply 1-2.This research mainly used CFRP hub and steel hub of the grinding wheel to explore the performance of surface roughness in the carbide materials (WC). The experimental results show the performance of the CFRP hub grinding wheel is better than the steel hub.
1-3. In Figure 5 Please make the amplitude scale same for better comparison.
Reply 1-3. We have revise them in Figure 5.

Reviewer 2 Report (New Reviewer)
Comments and Suggestions for Authors
There are some corrections to be made, as noted in the paper.
It is definitory to mention here that WC (tungsten carbide) is not a STEEL !
Use a metallurgist for understand this.

In text, as in the References, almost always appear " .; " after authors name. It mandatory to erase this group of symbols.
These symbols are the result of Copy/Paste operations.
Author Response
Review 2 comments and suggestions for authors
2-1.There are some corrections to be made, as noted in the paper. It is definitory to mention here that WC (tungsten carbide) is not a STEEL ! Use a metallurgist for understand this. Metallurgy teach us the STEEL is an alloy of Fe and C and others elements. even W ! But WC is not STEEL !
Reply 2-1.We have finished the corrections all in the paper. Include the tungsten steel modify to the tungsten carbide.

Reviewer 3 Report (New Reviewer)
Comments and Suggestions for Authors
The article entitled “An investigation of grinding WC cemented carbide with different diamond wheel hubs” by Lin et al. proposes the use of CFRP hub in replacing steel during grinding of tungsten. The article has 2 tables, 8 figures and 18 references. The authors compared the performance of steel and CRPF grinding wheel, through empirical and experimental methods. Though the work attempted is appreciated, the prevailing issues must be addressed before being considered for publication. Few of them were given below for author’s consideration.
1.Title does not convey the work attempted. Use of short forms in the title is not recommended.
2. Authors are directed to mention author names according to Journal format. Use of last name of the first author followed by et al. is recommended.
3. Authors are expected to quote what is a hardness-brittle material?
4. Author should prune careless mistakes prevailing in the manuscript. Page 2 para 2 line 6 from top- word fever?
5. According to the literature review, uses of CRPF hub grinding wheel are reported by earlier researchers. Authors need to comply with the need for the current study. Novelty of the present study is major concern.
6. Flow of statements is discontinuous in the whole manuscript.
7. The quality of figures is not adequate for a research article. The content of Figs.6-8 is not readable.
8. Acronyms should be expanded during the first time of usage ex. Page 5 line 3 from top GC
9. Basic definitions such as damping effect should not be given in a research article.
10. Is Fig.4 schematic diagram or photograph?
11. Why higher, middle and lower frequency were fixed as 5500, 500 ad 300 Hz? A huge variation prevails between higher and middle range.
12.Discussion section is disappointing. The results should be analyzed by comparing their results with published literature.
13. More relevant literatures should be added.
Comments on the Quality of English LanguageSignificant improvement required.
Author Response
3-1.Title does not convey the work attempted. Use of short forms in the title is not recommended.
Reply 3-1.We have revise the title. The new title is “The Study of the Performance of the Diamond Wheel’s Steel and CFRP Hubs in Grinding Tungsten Carbide (WC)”.
3-2. Authors are directed to mention author names according to Journal format. Use of last name of the first author followed by et al. is recommended.
Reply 3-2.We have revise them as quote and reference.
3-3. Authors are expected to quote what is a hardness-brittle material?
Reply 3-3.We add in the para 1 line 1 and para 2 line 2 of the introduction as reference [1], [2], [7], and [8].
3-4. Author should prune careless mistakes prevailing in the manuscript. Page 2 para 2 line 6 from top-word fever?
Reply 3-4. We have revise the mistakes.
3-5. According to the literature review, uses of CRPF hub grinding wheel are reported by earlier researchers. Authors need to comply with the need for the current study. Novelty of the present study is major concern.
Reply 3-5. This research mainly used CFRP hub and steel hub of the grinding wheel to compare the performance in the surface roughness for the carbide materials(WC). As WC is one of hardness-brittle material. It must use the novel hub to increase the performance in the surface roughness. Steel hub is the traditional grinding wheel. The experimental results show the performance of the CFRP hub is better than the steel hub in the grinding WC materials.
3-6. Flow of statements is discontinuous in the whole manuscript.
Reply 3-6.We have revise the content in the whole manuscript.
3-7. The quality of figures is not adequate for a research article. The content of Figs.6-8 is not readable.
Reply 3-7.We have improve the quality of figures in Figs.6-8.
3-8. Acronyms should be expanded during the first time of usage ex. Page 5 line 3 from top GC
Reply 3-8. GC is one of the specification in the dressing block. We have revise dressing block(GC-220).
3-9. Basic definitions such as damping effect should not be given in a research article.
Reply 3-9.We have delete the equations in (4) to (6).
3-10. Is Fig.4 schematic diagram or photograph?
Reply 3-10.Fig.4 is the schematic diagram for oscillation test in the CFRP hub and steel hub of the grinding wheel.
3-11. Why higher, middle and lower frequency were fixed as 5500, 500 ad 300 Hz? A huge variation prevails between higher and middle range.
Reply 3-11.The purpose is to distinguish the damping difference in the lower and higher frequency.
3-12.Discussion section is disappointing. The results should be analyzed by comparing their results with published literature.
Reply 3-12. This research mainly used CFRP hub and steel hub of the grinding wheel to explore the performance of surface roughness in the carbide materials (WC). So that we use the [13] & [17] of the reference to compare the difference in the discussion section.
3-13. More relevant literatures should be added.
Reply 3-13.We have add more relevant literatures in the reference, quote [1], [2], [7], [8] and explain the difference with [13] & [17] in the section of the discussion.

Reviewer 4 Report (New Reviewer)
Comments and Suggestions for Authors
I made the following comments to increase the research quality:
1. In the paper the authors are proposing to develop a carbon fiber-reinforced thermoplastic (CFRP) hub for high-speed grinding of Tungsten steel, based on a comparison with the classical hub made from steel. I am considering that the topic is new in the field and the number of publications ale still limited.
Recommendation 1: Novelty must be highlighted in a better way.
2. Chapter 2 presents the experimental method. In 2.1. the authors are made a synthesis about the grinding principles.
Recommendation 2: 2.1. A scheme about the grinding forces is needed.
2.2. In figure 1, (b) is used two times.
In 2.2. the authors present the main characteristics of the experimental equipment’s. Conventional and modern equipment are used. A detailed analysis of the grinding wheels material is made.
Recommendation 3: 3.1. Please explain how the grinding material of the wheel is fixed on the
CFRP hub
In 2.3. the authors present the method for measuring in the static condition of the damping effecr for the two types of wheels.
Recommendation 4: What is the impact hammer hardness’s.
In 2.4. the authors present the methodology for measuring the surface quality obtained by using the steel and CFRP hub.
3. Chapter 3 presents the experimental results in terms of oscillation of the CFRP hub and steel hub grinding wheel in the grinding process and surface roughness and morphology of WC workpiece after grinding. The results are clearly presented, and the figures comes to validate the results.
4. Chapter 4 is dedicated to the discussions about the results of the experimental work. These discussions relived the advantages of the new type of hub wheels for high-speed grinding and are made critical in comparison with the other research.
5. The conclusions are in accordance with the experimental work and the future work is well highlighted.
The references are appropriate with the subject.
The paper could be published with minor revisions.
Author Response
4-1. In the paper the authors are proposing to develop a carbon fiber-reinforced thermoplastic (CFRP) hub for high-speed grinding of Tungsten steel, based on a comparison with the classical hub made from steel. I am considering that the topic is new in the field and the number of publications ale still limited.
Recommendation 1: Novelty must be highlighted in a better way.
Reply 4-1. Thank you for reviewer approve. This research mainly used CFRP hub and steel hub of the grinding wheel to explore the performance of surface roughness in the carbide materials (WC). The experimental results show the performance of the CFRP hub grinding wheel is better than the steel hub.
4-2. Chapter 2 presents the experimental method. In 2.1. the authors are made a synthesis about the grinding principles.
Recommendation 2: 2.1. A scheme about the grinding forces is needed.
2.2. In figure 1, (b) is used two times.
In 2.2. the authors present the main characteristics of the experimental equipment’s. Conventional and modern equipment are used. A detailed analysis of the grinding wheels material is made.
Recommendation 3: 3.1. Please explain how the grinding material of the wheel is fixed on the CFRP hub
3-2. Recommendation 3 reply: We manufactured the diameter of Ø205*t10mm to fixed in the CFRP hub of diameter Ø205 with resin adhesive to do the binders.
In 2.3. the authors present the method for measuring in the static condition of the damping effect for the two types of wheels.
Recommendation 4: What is the impact hammer hardness’s.
4-2. Recommendation 4 reply: We use PCB piezotronics TLD086C01_084C05 hammer to impact the grinding wheel. The specifications were not explain the hardness. But it illustrate the hammer’s tip is soft.
In 2.4. the authors present the methodology for measuring the surface quality obtained by using the steel and CFRP hub.
- Chapter 3presents the experimental results in terms of oscillation of the CFRP hub and steel hub grinding wheel in the grinding process and surface roughness and morphology of WC workpiece after grinding. The results are clearly presented, and the figures comes to validate the results.
- Chapter 4 is dedicated to the discussions about the results of the experimental work. These discussions relived the advantages of the new type of hub wheels for high-speed grinding and are made critical in comparison with the other research.
- The conclusions are in accordance with the experimental work and the future work is well highlighted.
The references are appropriate with the subject.
The paper could be published with minor revisions.
4-2. Recommendation 5 reply:Thank you for reviewer approve and we have revise the content in the whole manuscript again.

Round 2
Reviewer 3 Report (New Reviewer)
Comments and Suggestions for Authors
Though the authors are appreciated for carrying out a revision, the revision is not satisfactory. Authors failed to address most of the queries raised by the reviewer.
1. The abbreviation Ra and Rz are not presented in the abstract section.
2. The flow of statements are still discontinuous. Language is not adequate for a research article, reaching global audience.
3. Authors have mentioned that use of CRPF hubs are in existence in the introduction section. What is novel in the present study
4. The content inside the figures are still not readable.
5. There is a big scope for improving the discussion section.
Comments on the Quality of English LanguageExtensive English editing is mandatory
Author Response
- The abbreviation Ra and Rz are not presented in the abstract section.
Reply 3-1. We have illustrated in the line 12 & 13 of the abstract section.
- The flow of statements are still discontinuous. Language is not adequate for a research article, reaching global audience.
Reply 3-2. The resubmitted manuscript have undergo extensive English revisions by a native English people to correction as the attachment file of the English revisions proof.
- Authors have mentioned that use of CRPF hubs are in existence in the introduction section. What is novel in the present study.
Reply 3-3. This paper present a carbon fiber-reinforced thermoplastic (CFRP) hub for high-speed grinding of Tungsten steel based on a comparison with the classical hub made from steel. The experimental results shown the performance is better than the traditional steel hub. As the topic is new in the grinding field and the number of publications are still limited. However, we also add description in the paragraph 4 of the introduction.
- The content inside the figures are still not readable.
Reply 3-4. We have revised the figures’ qualitity in Figure 5 ~ 8 in the resubmit manuscript.
- There is a big scope for improving the discussion section.
Reply 3-5. We have improve the discussion section as yellow highlight in page 11 and 12.

Round 3
Reviewer 3 Report (New Reviewer)
Comments and Suggestions for Authors
The article is recommended for publication
Comments on the Quality of English LanguageLanguage demands improvement.
This manuscript is a resubmission of an earlier submission. The following is a list of the peer review reports and author responses from that submission.
Round 1
Reviewer 1 Report
Comments and Suggestions for Authors
Reviewer: Minor revision
This paper reports a detailed study about the exploring how to reduce the oscillation and improve processing efficiency and morphology in the grinding process at different peripheral speeds of the grinding wheel. The proposed approach is very meaningful, and the manuscript organization is satisfied. So, I think that this paper deserves to be published in journal of Applied Sciences after minor revision of some issues as follows:
1- Authors must ensure that the quality of English is improved (i.e., make all efforts to rectify any grammatical mistakes, typos, double spaces, missing spaces etc.).
2- Authors must support all equations throughout the manuscript with suitable references.
3- The figures need to be clearer. The authors must improve the quality of the figures (For example figs. 4&5).
4- For simplicity, the authors recommended to collect all data in a clear table.
5- In the discussion part, authors must give an explanation about the effect of thermal generation on the grinding speed.
6- Authors must support the conclusion with some technical results.
7- It will be better to compare the performance in this work with previously reported works. Authors must use suitable recent references.
Comments on the Quality of English LanguageMinor editing of English language required.
Author Response
Reply of question 1: We have check the full paper and modify all the related issues.
Reply of question 2 : We have add relevant references in all equations. For example [11], [15], [16], [23] and [24].
Reply of question 3 : We have improve the qualities of the figures as in Figure 4 and Figure 5.
Reply of question 4 : We have add in the Table 3 of the section in the discussion.
Reply of question 5 : This paper explore how to use novel material(CFRP) to replace the steel hub to improve the grinding qualities and the surface morphology in the hardness and brittle of cemented carbide. Thermal generation is another issue on the grinding speed. Maybe we will study in the future.
Reply of question 6 : The grinding qualities and the surface morphology of the CFRP hub is better than the steel hub when cemented carbide (WC) was grinded at the same peripheral speed of the diamond grinding wheel as Figure 5 and Figure 6 in this paper. In addition, the material properties of CFRP hub is better than the steel hub in the elastic modulus and thermal expansion coefficient as Table 1. Therefore, these technical results can be able to support the conclusion.
Reply of question 7 : Thank you reviewer suggestion. We use references in recent five years as in [1], [2], [13], [16] and [19].

Reviewer 2 Report
Comments and Suggestions for Authors
1. The results and discussion section is extremely poor in quality of discussion/explanations. About 75% of the manuscript (1500 words out of 5600) consists of introduction, description of basic methods such as roughness measurements and reference list. The authors have filled the experimental section with graphs images touching just tangentially the subject, without significant scientific explanations.
Just the introduction chapter is bigger than the whole experimental section. This speaks clearly about the scientific soundness of present work.
2. Amplitude is bigger and the stabilisation of oscilation is poorer for low frequency vibrations for the CFRP hub. The authors should study what are the freqeuencies when they grind and see wheter they are high frequency or low frequency.
Only at large 5500 Hz an improvement in stabilisation can be noticed for the CFRP hub.
What is the precision for the method used for roughness measurement, Cg and CgK parameters for such an method?
3. Authors state :"WC is a compound composed of WU and carbide. " What is WU? Tungsten??? "This is because WV is hard and brittle" Maybe WC.
4. The two grinding wheels presented in Figure 3 appear to have different grinding material. This aspect puts a great question mark on whether the observed results are due to different materials or different hubs. Authors should clarify this aspect and characterise the grinding wheels also.
Author Response
Reply the question of No 1. This paper is to discuss how to use novel hub of the grinding wheel to improve the grinding qualities in the hard and brittle WC materials. For example the surface roughness and the morphology. The traditional hub of the grinding wheel is manufactured by steel. We use CFRP hub to improve the surface roughness and the morphology. The scientific explains as an Table 1 and Table 3.
Reply the question of No 2. : The grinding is near to the high frequency when the peripheral speed of the grinding wheel at 60-80m/s.
We use the equipment of ZYGO PERFORMA-NCENewView8300 to measure the roughness of the WC workpiece. We didn’t measurement the parameters of Cg and CgK. We did repeated measurement in the highness of the SiC workpiece. The number of measurements are thirty order. The error range of repeated measures is below to 0.1% compare to the standard value. For example the roughness of the surface is 1.8μm. The error of the measurement is ±0.0135μm. This information is offered by ZYGO Corp..
Reply the question of No 3. : We have revised “Tungsten carbide is a compound composed of WC and Co”. The chemical symbol of the tungsten carbide is WC. In addition, we also revised WV to WC.
Reply the question of No 4. : The grinding materials are the same as the file.

Round 2
Reviewer 2 Report
Comments and Suggestions for Authors
1. The results and discussion section is extremely poor in quality of discussion/explanations. About 75% of the manuscript (1500 words out of 5600 is the original part) consists of introduction, description of basic methods such as roughness measurements and reference list. The authors have filled the experimental section with graphs images touching just tangentially the subject, without significant scientific explanations.
Just the introduction chapter is bigger than the whole experimental section. This speaks clearly about the scientific soundness of present work.
This issue hasn't been adressed and the original part (results and discussions of these results is just 1500 words out of a manuscript containing 5600 words. Authors say that the paper is to discuss the novel hub, but they don't discuss the data and provide explanations for the observed behaviour.
2. Amplitude is bigger and the stabilisation of oscilation is poorer for low frequency vibrations for the CFRP hub. The authors should study what are the freqeuencies when they grind and see wheter they are high frequency or low frequency.
Only at large 5500 Hz an improvement in stabilisation can be noticed for the CFRP hub. What is the precision for the method used for roughness measurement, Cg and CgK parameters for such an method?
Cg and Cgk still missing for such a method. The specifications of the used equipment and theoretical precision are only on standard sample and controlled conditions (as stated by manufacturer). At high frequency, the amplitude is significantly lower for steel hub (25%), however due to misrepresentation of data (different y axis) the graphic leads to false impresion.
3. Authors state :"WC is a compound composed of WU and carbide. " What is WU? Tungsten??? "This is because WV is hard and brittle" Maybe WC.
Authors have addresed this issue.
4. The two grinding wheels presented in Figure 3 appear to have different grinding material. This aspect puts a great question mark on whether the observed results are due to different materials or different hubs. Authors should clarify this aspect and characterise the grinding wheels also.
The images provided in the response suggest similar materials, however, the images from manuscript, due to different white balance, indicate different materials. Please ammend this. The table 3, does not constitute a characterisation of the grinding wheels. Weight light or heavy, or elastic modulus low or high is not scientific at all. Please improve the quality of the discussions/explanations.
Author Response
Reply 1. We have offer the experimental data and provide explanations in the section of 5.1 and 5.2. And then, showed the experimental data in Figure 4 and Figure 5 to compare the advantage in the CFRP hub and steel hub.
Reply 2. Cg is 1.47. At high frequency, the amplitude is significantly lower for steel hub (25%) compare to the CFRP hub. CFRP hub can be able to reduce the chatter vibration in the short time. This is due to the material of the WC workpiece is hardness so that the soft material (CFRP hub) touch workpiece instantly prodcue higher amplitude. As the weight of the CFRP hub is slightly and has lower elastic modulus to achieve the function of damping.
Reply 4. We have ammend the Figure 3 and discussions. Damping is the key point to reduce the chatter vibration and increase the finessing in the grinding.
